# 10-DEBC Hydrochloride as a Promising New Agent against Infection of *Mycobacterium abscessus*

**DOI:** 10.3390/ijms23020591

**Published:** 2022-01-06

**Authors:** Da-Gyum Lee, Hye-Jung Kim, Youngsun Lee, Jung-Hyun Kim, Yoohyun Hwang, Jeongyeop Ha, Sungweon Ryoo

**Affiliations:** 1Center for Clinical Research, Masan National Tuberculosis Hospital, Changwon 51755, Korea; sh050301@naver.com (D.-G.L.); hwangyoohyun@gmail.com (Y.H.); 2New Drug Development Center, KBIO OSONG Medical Innovation Foundation, Cheongju 28160, Korea; hjkim@kbiohealth.kr (H.-J.K.); myid15@kbiohealth.kr (J.H.); 3Division of Intractable Diseases Research, Department of Chronic Diseases Convergence Research, Korea National Institute of Health, Cheongju 28160, Korea; iadoj@korea.kr (Y.L.); kjhcorea@korea.kr (J.-H.K.)

**Keywords:** *Mycobacterium abscessus*, 10-DEBC, drug resistance, nonreplicating condition, surrogate caseum binding assay

## Abstract

*Mycobacterium abscessus* (*M. abscessus*) causes chronic pulmonary infections. Its resistance to current antimicrobial drugs makes it the most difficult non-tuberculous mycobacteria (NTM) to treat with a treatment success rate of 45.6%. Therefore, there is a need for new therapeutic agents against *M. abscessus*. We identified 10-DEBC hydrochloride (10-DEBC), a selective AKT inhibitor that exhibits inhibitory activity against *M. abscessus*. To evaluate the potential of 10-DEBC as a treatment for lung disease caused by *M. abscessus*, we measured its effectiveness in vitro. We established the intracellular activity of 10-DEBC against *M. abscessus* in human macrophages and human embryonic cell-derived macrophages (iMACs). 10-DEBC significantly inhibited the growth of wild-type *M. abscessus* and clinical isolates and clarithromycin (CLR)-resistant *M. abscessus* strains. 10-DEBC’s drug efficacy did not have cytotoxicity in the infected macrophages. In addition, 10-DEBC operates under anaerobic conditions without replication as well as in the presence of biofilms. The alternative caseum binding assay is a unique tool for evaluating drug efficacy against slow and nonreplicating bacilli in their native caseum media. In the surrogate caseum, the mean undiluted fraction unbound (*fu*) for 10-DEBC is 5.696. The results of an in vitro study on the activity of *M. abscessus* suggest that 10-DEBC is a potential new drug for treating *M. abscessus* infections.

## 1. Introduction

NTM is ubiquitous in soil and drinking water sources [1]. NTM often results in high person-to-person contact rates with pathogens [1,2]. Infections caused by NTM are increasing rapidly, and in developed countries with abundant resources, these infections surpass tuberculosis [3]. Factors contributing to the increased incidence of NTM infection include an increase in susceptible host populations, such as the elderly and those with cellular immune system defects due to disease or immunosuppressive therapy [3,4].

*M. abscessus* is one of the most common NTMs in pulmonary infections that can cause lung or other diseases in healthy individuals. The number of infections affecting the extrapulmonary part of the body is also increasing [3,5,6].

*M. abscessus* belongs to Runyon group IV, classified as rapid-growing mycobacteria (RGM). It has been reported that RGM in Japan and America causes 5% of pulmonary infections, and the 65%–80% of the RGM infections are caused by *M. abscessus* [2]. Pulmonary disease caused by *M. abscessus* is common in patients with cystic fibrosis (CF), while disseminated infections occur in immunocompromised patients. Patients with CF have an increased risk of disease with *M. abscessus* [7]. Prospective studies have shown that CF is one of the major risk factors for NTM infection, and along with *M. abscessus* is the most common by far [8,9]. In CF, *M. abscessus* causes rapid inflammatory lung injury [10], often untreatable, and precludes safe lung transplantation, leading to poor clinical outcomes or lifelong asymptomatic infection [11,12,13,14].

The alarming fact is that *M. abscessus* is the most common solitary lung infection, with an average treatment success rate of only 45.6% [15,16]. The specific treatment success rate was 56.7% for *M. abscessus* subsp. *Massiliense* [15]. *M. abscessus* is one of the most resistant microorganisms to chemotherapeutic therapy, and its main threat as a human pathogen is its high resistance to multi-drugs. *M. abscessus* is also associated with biofilm formation, germicide resistance, high temperatures, and acidic environments [17]. For patients with chronic lung disease due to NTM, none of the currently available treatments have been proven effective for long-term sputum conversion [15]. The high resistant rate of *M. abscessus* may even disable antibiotic combination therapy. 

Current regimens recommend drug combinations for the treatment of *M. abscessus* pulmonary infections. The American Thoracic Society and the Infectious Disease Society of America recommend CLR, amikacin (AMK), azithromycin (AZR), cephalosporin (a type of β-lactam), cefoxitin (FOX), and carbapenem (another type of β-lactam) imipenem (IPM). Still, less than half of patients with *M. abscessus* infection can be cured with these treatments [4,18]. Consequently, identifying novel treatment approaches is imperative. There is increasing research on experimental antibiotics that have potential activity against *M. abscessus* through various mechanisms and new therapies. However, there is not enough research on its effectiveness against *M. abscessus*. 

10-DEBC is a selective Akt inhibitor (also known as PKB) [19]. Akt is a serine/threonine kinase. It phosphorylates and inactivates components of the apoptotic mechanism, including caspase 9 and BCL2-associated agonist of cell death (BAD). Akt phosphorylates and inhibits the forkhead box’s transcription factor, thus promoting cell survival [20]. 

Han et al. identified a novel compound called 10-DEBC [21]. However, the mechanism of 10-DEBC in *M. abscessus* is not yet well understood. This study demonstrated that 10-DEBC exhibited activity against *M. abscessus*, both in vitro and intracellular. 

## 2. Results

### 2.1. 10-DEBC Is Active against M. abscessus

The Clinical & Laboratory Standards Institute (CLSI) currently recommends using the Mueller–Hinton (MH) microdilution broth-based method for determining the MIC of antimicrobial agents for RGM [22]. Therefore, we obtained MIC of 10-DEBC for *M. abscessus* in MH broth with resazurin rather than the 7H9 broth-based method. 

The IC_50_ of 10-DEBC for *M. abscessus* was 3.006 μg/mL in MH broth and 5.81 μg/mL in 7H9 broth, respectively (Appendix A). The MIC_90_ of 10-DEBC for *M. abscessus* was 4.766 μg/mL in MH broth and 9.53 μg/mL in 7H9 broth, respectively (Appendix A). 10-DEBC exercised similar powers to reference strains of *M. abscessus*, regardless of the type of culture medium MH or Middlebrook 7H9. As shown in Figure 1A, 10-DEBC showed excellent in vitro activity comparable to other reference compounds, such as CLR, AMK, and streptomycin (STR). They have different mechanisms of action.

In our study, the in vitro IC_50_ of CLR, AZR, AMK, and STR against *M. abscessus* was 0.613, 5.784, 1.34, and 18.52 μg/mL, respectively. The MIC_90_ of CLR, AZR, AMK, and STR against *M. abscessus* was 1.869, 39.251, 3.125, and 36.34 μg/mL, respectively.

We have adapted reporter-based assays well suited for drug discovery applications because of their simplicity and sensitivity compared with dye and absorbance-based assays [23]. We constructed luminescence reporter strains and then used them to determine the MICs of CLR and 10-DEBC. First, the in vitro activities of 10-DEBC and CLR were measured by reporter-based bioluminescence; then, MIC data were collected using the REMA method. We used bioluminescent strains of *M. abscessus* to confirm our drug susceptibility. The dose–response curves (DRC) of 10-DEBC in *M. abscessus*-LuxG13 are shown in Figure 1B, and the IC_50_ value is 3.9 μg/mL and MIC_90_ value is 12.5 μg/mL. Therefore, 10-DEBC can be considered an effective drug candidate for *M. abscessus*.

### 2.2. 10-DEBC Is Active against Clinical Isolates of the M. abscessus and Clarithromycin Resistant Mutant

We next determined whether 10-DEBC retained this potent activity compared with clinical isolates of *M. abscessus*, including Rough (R) and Smooth (S) colony types. 10-DEBC was also inhibited against nine strains in the clinical isolates, with IC_50_ from 1.69 to 2.63 μg/mL and MIC_90_ from 2.38 to 4.77 μg/mL, similar to the reference strains of the subspecies (Table 1). The R phenotype tends to be much stronger than the S type in combating the host defense mechanism [24,25]. These results indicate that 10-DEBC is effective in vitro against the reference strain and clinically isolated R and S.

CLR is a key agent in the treatment of *M. abscessus* complex infections. The main cause of treatment failure is clarithromycin inducible resistance [26]. From this study, we tested whether 10-DEBC effectively inhibited the growth of drug-resistant strains that were laboratory generated from this study at high concentrations (100 Please consider this suggested change and same as follows.) of CLR. As shown in Figure 2, the laboratory-generated resistance mutant showed high resistance levels to the CLR. 

In addition, the anti-CLR variant was sensitive to 10-DEBC as an IC_50_ band (4.662~5.535 μg/mL) and MIC_90_ 12.5 μg/mL. It was the same regardless of whether it was a CLR-resistant or a susceptible strain. Thus, 10-DEBC should be an active drug to treat CLR-resistant *M. abscessus*. These results confirmed that 10-DEBC was active against clarithromycin-sensitive and -resistant strains. Therefore, 10-DEBC should be viewed as an active drug candidate for the treatment of clarithromycin-resistant *M. abscessus*.

### 2.3. 10-DEBC Is Susceptible to Nonreplicating and Biofilm Growing M. abscessus

The activity of 10-DEBC in nonreplicating phase cultures was examined, and this phase, that was induced by lack of oxygen, was measured. Nonreplicating status was confirmed by measuring the growth curves of *M. abscessus* under aerobic and anaerobic conditions before assessing the drug action (data not shown) [27]. 

CLR showed increased activity against anaerobic cultured *M. abscessus* (IC_50_ 1.309 μg/mL and MIC_90_ < 10 μg/mL) than aerobic condition (IC_50_ 0.034 μg/mL and MIC_90_ 0.078 μg/mL), demonstrating a significant 37.52-fold and 128.21-fold shift to higher IC_50_ values and MIC_90_ values (Figure 3). Interestingly, in an anaerobic state, the IC_50_ value of 10-DEBC increased 1.18 times, and MIC_90_ of 10-DEBC increased 2 times compared to the aerobic culture, showing similar results to the aerobic culture. Thus, 10-DEBC attained activity against anaerobic *M. abscessus* closely related to the nonreplicating environment. 

Biofilm growth is a necessary factor in antibiotic resistance. Resistance to antibiotics, disinfectants, and biocides by biofilm-forming microorganisms can lead to treatment failure. Clinical experience has shown that biofilms must be removed to resolve the infection [28]. According to Muñoz-Egea, differences were found between MICs and minimum biofilm eradication concentration (MBEC) in *M. abscessus*; specifically, the MBEC of the CLR increased 100,000 times compared with the MIC [28]. However, when *M. abscessus* was exposed to CLR, MBEC was 6000-fold higher from 0.034 to >200 μg than the typical MIC of CLR.

As can be seen in Figure 4, CLR has excellent bactericidal activity (IC_50_ = 0.034 μg/mL) in growing culture but lost its activity ultimately against biofilm-growing *M. abscessus* (IC_50_ > 200 μg/mL). On the other side, 10-DEBC still has an IC_50_ of 38.72 μg/mL and MIC_90_ of 50 μg/mL against biofilm-growing bacteria. It seems to be active against *M. abscessus* that grows on biofilms. These results suggest 10-DEBC is an attractive substance for all types of anti-*M. abscessus* treatment. 

### 2.4. 10-DEBC Is Effective against Intracellular M. abscessus

We assessed the cell viability at various concentrations to determine whether 10-DEBC affects cytotoxicity. 10-DEBC has not observed the cytotoxicity under particular concentrations that inhibited intracellular *M. abscessus* (data not shown) and did not show significant toxicity against macrophages differentiated from primary stem cells [21]. According to Janjetovic et al., as shown by LDH and MTT tests, 10-DEBC dihydrochloride at a concentration of 10 μM did not affect cell viability when applied alone [19].

The dual reading test made it possible to obtain DRC for *M. abscessus* inhibition and cytotoxicity of THP1 cells [29]. IC_50_ and CC_50_ can be defined (cell toxicity concentration that induces 50% cell death) in this study. Figure 5 shows the dual DRC to 10-DEBC and the data obtained from the same screening wells over time. Cytotoxicity was measured with the SYTO 60 probe. Treatment with CLR successfully rescued cells from infection challenge and prevented bacterial replication (data not shown). Kim et al. showed that CLR significantly reduced the intracellular *M. abscessus* present two days after infection at concentrations of 0.1 μg/mL [30]. 10-DEBC inhibited the growth of *M. abscessus* (lux) in a dual read assay with an IC_50_ of 3.48 μg/mL, with no cytotoxicity for THP1, even at the highest concentrations (Figure 5A). Consistently with the THP-1 infection model, the intracellular inhibitory activity of 10-DEBC was shown (Figure 5B). 10-DEBC was found to inhibit the intracellular growth of *M. abscessus* in THP-1 cells with an IC_50_ of 13.18 μg/mL (Figure 5B), while inhibiting replication of *M. abscessus* inside human embryonic cell-derived macrophages (iMACs) with an IC_50_ of 5.8 μg/mL, without host cell toxicity even at the highest concentration of 50 μg/mL (Figure 5C,D).

### 2.5. 10-DEBC Is Penetrated in Caseum Surrogate

The caseum surrogate binding assay is a valuable tool in drug discovery for tuberculosis and NTM and is used in many lead optimization programs. The higher the *fu*, the more the drug diffuses into caseum. To avoid collecting caseum from infected animals, we designed a caseum simulation that will mimic the binding properties of a real matrix. 

THP-1 monocytes were exposed to 94 μg/mL oleic acid and 94 μg/mL linoleic acid, which induce differentiation into foamy macrophage (FM) in vitro [31]. PMA-differentiated THP-1 macrophages (THPM) accumulated lipid droplets most significantly in the presence of 94 μg/mL oleic acid and 94 μg/mL linoleic acid. THPMs were treated with oleic acid and linoleic acid for 24 h, and the resulting FM were infected, washed, harvested, lysed, and denatured before being used in the RED assay. 

The drug concentration in the donor and receiver chamber of the RED assay was quantified by LC-MS. KBIO (Osong, Korea) conducted this experiment. Drug quantification was achieved using liquid chromatography coupled to mass spectrometry (LC-MS) methods (Appendix A). The LC-MRM strategy was used for quantification of drugs in the donor and receiver chamber of RED. Respective parameters determined for analytes used herein are shown in Appendix A.

The fraction unbound of CLR and 10-DEBC were determined in matrixes with triplicate samples for each batch (Table 2). The mean *fu* (%) for CLR was 39.92% in the surrogate matrix, and undiluted *fu* was 86.91. The 10-DEBC’s average caseum-free fraction (caseum *fu*%) of 0.04% to 0.06% (standard deviation (SD), 0.005%) and undiluted *fu* of 5.696 was much lower than CLR’s caseum fraction. 

## 3. Discussion

Pulmonary NTM is a recognized disease in developed countries such as Japan [32] and USA [33]. Opposite trends in NTM and tuberculosis rates were observed in 75% of 16 regions on four continents [34]. When talking about diseases caused by NTM, it remains tough and especially involves *M. abscessus* [35,36]. It is challenging to treat *M. abscessus* like other NTMs; it is resistant to most antibiotics, including macrolides, aminoglycosides, rifamycins, tetracyclines, and β-lactams. There is no consistent regimen with demonstrated or predicted efficacy for pulmonary disease caused by *M. abscessus*. Unfortunately, despite combined treatment, success rates only range from 25% to 42% [37]. Approximately 20% of *M. abscessus* strains isolated from pulmonary infections do not respond to the macrolides included in treatment [38,39]. Therefore, there is an urgent medical need to discover and develop new and more effective anti-*M. abscessus* drugs.

It is already known that the R phenotype is much more virulent than S morphology and tends to cause very persistent and difficult-to-treat infections [25,40]. The R form is involved in chronic airway colonization in CF [24]. 10-DEBC is effective in vitro against both the reference strain and the clinically isolated strains of R and S colony morphologies in our study. In this regard, 10-DEBC has additional clinical benefits.

CLR is an essential antibiotic for treating *M. abscessus* infections, but intrinsic and acquired resistance to CLR make treatment increasingly tricky and have unsatisfactory outcomes [41]. This is common in clinical isolates of *M. abscessus*, especially clinical R-type strains. We used spontaneous CLR-resistant mutants to test whether 10-DEBC is effective against CLR-resistant strains. Additionally, these CLR-resistant mutants show high drug resistance to CLR. Our in vitro experimental results, as shown in Figure 2 and Table 1, suggest that 10-DEBC is functioning to susceptible, induced CLR-resistant, intrinsic CLR-resistant, and clinical R morphotype *M. abscessus* strains. Our experimental results demonstrate that 10-DEBC has a suggestive advantage in being used as a core drug candidate for *M. abscessus* disease.

In a recent transcriptomic study, the total *M. abscessus DosR* regulon increased under hypoxia [42]. *DosR*, a modulator of the resting survival response necessary for the survival of tubercle bacilli under hypoxia, is maintained in all NTM species [43]. The ability of *M. abscessus* to generate biofilm represents a successful survival strategy for these ubiquitous microorganisms. They form a biofilm on the airways’ surface inside the human lung [44]. 

It suggests that *M. abscessus* have the conserved molecular strategies for persisting inside the host. As the disease evolves, *M. abscessus* may not replicate and exhibit phenotypic drug resistance within lung lesions, which may be one of the factors contributing to the persistence of *M. abscessus* infection during long-term treatment [45,46]. This behavior is related to their pathogenicity and increased antibiotic resistance [47,48]. In general, the drug effort is established against *M. abscessus* active during aerobic growth. Consequently, treatment outcomes vary with the use of the same antibiotics against different bacilli phenotypes, and they are often ineffective. Most *M. abscessus* remain nonrenewable and are in a low metabolic state in granulomas and biofilm formation in the pulmonary airways. Situations in the host are entirely different from in vitro testing for conventional antibiotic activities. In this study, we examined the efficacy of 10-DEBC against different phenotypes of *M. abscessus*, which survived in biofilm and anaerobic environments as found in the lungs of patients. The superiority of 10-DEBC is that it is effective even under adverse anoxic and biofilm conditions.

NTMs can grow and survive outside and within cells, such as inside macrophages. In pulmonary infection with NTM, the bacilli attract the mucosa and are phagocytosed by macrophages. Infected macrophage cell lines may exhibit physiological conditions mimicking actual NTM disease with resistance to *M. abscessus*. As shown in Figure 5, 10-DEBC inhibited the growth of *M. abscessus* in macrophage and macrophage cell lines derived from human embryonic cells. 

The discovery of more effective anti-TB drugs requires a deeper understanding of the different mechanisms of drug penetration into the envelope of tuberculous lesions because granulomas and necrotic niches both contain caseum, which is difficult to sterilize [49]. Rapid equilibrium dialysis has been used to measure drug binding in a small amount of caseum as it has been validated with various tissue homogenates [50]. Mass balance was systematically included to exclude the nonspecific binding of drugs to the device.

A caseum assay might be challenging to develop for NTM drug screening because caseum-producing animal models are not well established yet. Alternately, we used the caseum surrogate assay designed from artificial, cell-culture-based caseum. We tested in the surrogate caseum. The very strongly bound (*fu* ≤ 1%) compounds were almost entirely retained in the chamber for no effective penetration into the caseum. Meanwhile, less bound compounds (1% < *fu* < 100%) can diffuse into the nucleus to varying degrees; the higher the *fu*, the more it penetrates the necrotic area.

As shown in Table 2, the unbound fraction of 10-DEBC is lower than CLR. Sarathy et al. [51] showed that bedaquiline and clofazimine are highly bound compounds (*fu* ≤ 0.01%). Bedaquiline and clofazimine are almost entirely trapped within the donor chamber and are not effective at permeating the caseum.

10-DEBC was first reported as a cell-permeable phenoxazine-derivative inhibitor of AKT kinase phosphorylation [52]. Activating the AKT1 signaling promotes intracellular *Mycobacterium tuberculosis* (Mtb) survival in macrophages through several mechanisms, including inhibition of phagosome–lysosome fusion [53]. It probably applies to *M. abscessus* as expected. 10-DEBC is a member of the chlorpromazine family of compounds, which were demonstrated to inhibit Mtb [54] without being Akt inhibitors themselves. 10-DEBC appears to be enriched by the macrophages and has been suggested to bind calmodulin and interfere with calcium transport [55]. Therefore, 10-DEBC may inhibit *M. abscessus* growth by interfering with not yet revealed mechanisms. Further research is needed to characterize the mechanisms by which 10-DEBC mediates intracellular and extracellular death of *M. abscessus*.

In conclusion, the results of this study on 10-DEBC hydrochloride activity suggest that this is a potent new drug candidate against *M. abscessus* infection.

## 4. Materials and Methods

### 4.1. Bacterial Strains and Culture Conditions

The *M. abscessus* strain (DSMZ 44196) was grown at 37 °C in Middlebrook 7H9 broth (BD, 27130) supplemented with 10% OADC (BD, 212240). We obtained the nine clinical strains from the National Culture Collection for Pathogens (NCCP). These strains were sent to the Seoul Clinical Laboratory (SCL) for conventional microbiological and biochemical tests. SCL provided the mycobacterial identification results. Except for 10-DEBC, MICs for nine drugs were formed with the broth microdilution method (SCL’s guideline, Yongin-si, Gyeonggi-do, Korea).

*M. abscessus* CLR-resistant mutant was prepared as previously described [27]. Recombinant *M. abscessus* expressing green fluorescence protein and luminescence protein for the macrophage infection were prepared as previously described [27]. The nonreplicating condition was previously described [27]. 10-[4′-(N, N-Diethylamino)butyl]-2-chlorophenoxazine (10-DEBC) was obtained from Tocris Bioscience (Bristol, UK). 

### 4.2. DRC Testing

The MICs were determined according to the CLSI guidelines. We treated 10-DEBC in actively growing *M. abscessus*, *M. abscessus*-pLUX G13, CLR-R mutant, and anaerobic cultured *M. abscessus*. 

10-DEBC was diluted two-fold in a serial dilution in 96-well plates with a total volume of 100 uL. It was then incubated for 84 h at 37 °C before adding 0.025% resazurin reagent. After overnight incubation, the fluorescence of resorufin (metabolite resazurin) was determined using Synergy H1 Hybrid Multi-Mode Reader (Bio-Tek, Winooski, VT, USA). IC_50_ values were calculated from the raw fluorescence data using Prism 5.0 software (GraphPad Inc., La Jolla, CA, USA). The experiments were carried out with triplicates.

### 4.3. Biofilm Assays

The MBEC assay was previously described [27]. The biofilm formation was conducted and the 96-peg lid was placed onto a 96-well. Next, we prepared 200 μL 10-DEBC solutions with the medium and then added them to each well of the 96-well plate, leaving them for four days. Following incubation, MBEC was quantified by performing the optimized resazurin reagents.

### 4.4. Intracellular Killing Assay

THP-1 cells were treated with a final concentration of 50 nM PMA (Sigma Aldrich, St. Louis, MO, USA) for two days. Infection with *M. abscessus* harboring green fluorescent (pTEC15) or luciferase was carried out at 37 °C for three hours at an MOI of 2:1. After three washes with PBS, cells were incubated containing 50 μg/L gentamycin for 30 min and washed again. Then, 200 uL RPMI 1640 media containing the indicated concentration of 10-DEBC was left for four days. The macrophages were stained with SYTO 60 (Invitrogen, Waltham, MA, USA) dye at a final concentration of 5 µM for 30 min at 37 °C. Luminescence was measured on day four using a Synergy H1 microplate reader (Bio-Tek, Winooski, VT, USA).

iMAC cells were previously described and provided kindly by Dr. Jung-Hyun Kim [21]. The iMAC cell’s infection method using *M. abscessus* harboring green fluorescent was the same as differentiated THP-1 cells. Live-cell images were captured by automated microscopy using Lionheart™ FX automated microscopy (Bio-Tek, Winooski, VT, USA). The Agilent BioTek Gen5^TM^ 3.05 software object feature enables the calculation of cells within the imaging field.

### 4.5. Generation of Surrogate Caseum

The THP-1 cells were seeded on a 150 mm dish, exposed to 100 nM PMA, and allowed to adhere. We used lipid body production in THP-1 monocytes to simulate caseum formation. This experiment was carried out overnight with 100 X Linoleic acid-Oleic acid-Albumin (LOA) (Sigma L9655), which is known to induce differentiation into foamy macrophage in vitro [31]. Infection with *M. abscessus* was carried out at an MOI of 5:1. After three washes with PBS, cells were treated 50 μg/L gentamycin for 30 min and washed again. They were then incubated with RPMI 1640 media, having LOA for one day. Infected foamy macrophages were lysed during three freezing−thaw cycles. A 30 min incubation denatured proteins at 75 °C to mimic alterations in protein structure and 3D conformation in the necrotic core of granulomas.

All binding assays were conducted using the RED device (Thermo Fisher Scientific, Waltham, MA, USA) and were previously described [56]. Using liquid chromatography coupled to mass spectrometry (LC-MS) methods, drug quantification was achieved. Mass analysis and detection were performed on a SCIEX (Framingham, MA, USA) 6500 Q-trap triple-quadrupole mass spectrometer equipped with a turbo ion-spray ionization source. The HPLC system was Agilent (Santa Clara, CA, USA) 1290 series with a degasser, binary pump, autosampler, and thermostat column compartment. Chromatographic separation for all compounds was achieved with an Agilent Poroshell (2.1 × 50 mm, 2.7 µm) column using gradient conditions. Exponential gradient elution (10 min) increased the mobile phase composition from 0 to 95% Sol B. The gradient was then decreased to 5% B to equilibrate the column for the next run. The compounds were directly injected with 2 μL of our prepared sample with Sol A (1 mM ammonium acetate in DW and 0.1% FA) and Sol B (1 mM ammonium acetate in methanol and 0.1% FA) into the analytical column without a trap column; the flow rate was set to 0.4 uL/min. Data collection and processing were performed with Mass Hunter Quantitative Analysis software. The unbound fractions were calculated using previous studies’ methods [57,58]. 

The *f*u in diluted surrogate caseum was calculated as the ratio between buffer chamber and sample chamber concentrations, as shown in Equation (1). A dilution factor of 10 (D = 10) was applied to calculate *f*u in undiluted surrogate caseum, as shown in Equation (2) [58].
(1)$$fu=\frac{\left[Receiver\;Concentation\right]}{\left[Donor\;Concentration\right]}$$
(2)$$undiluted\;fu=\frac{1/D}{\left(\left(\frac{1}{fu}\right)-1\right)+1/D}$$

### 4.6. Ethics

The Institutional Review Board approved all studies of Masan National Tuberculosis Hospital (IRB-398837-2018-E34, approved on 14 January 2019) and the Institutional Biosafety Committees (MTHIBC-19-01 and MTHIBC-21-11, approved on 26 February 2019 and 15 July 2021). The ethical use of hESC was approved by IRB of Korea Centers for Disease Control and Prevention: approval number 2020-02-09-C-A.

## Figures and Tables

**Figure 1 ijms-23-00591-f001:**
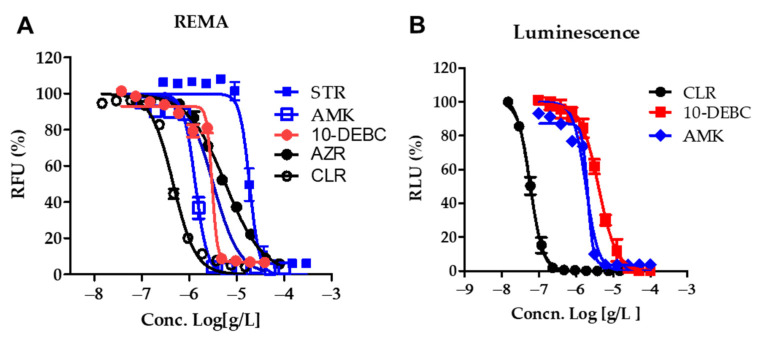
In vitro activity of 10-DEBC. (**A**) The activity of 10-DEBC against *M. abscessus* in MH broth medium. DRC was plotted from the REMA. RFU, relative fluorescence units. (**B**) The activity of 10-DEBC against *M. abscessus*-Lux, incubated under the same condition as in panel B RLU, relative luciferase units. The experiments were carried out with three biological replicates and expressed as the mean ± SEM for each concentration.

**Figure 2 ijms-23-00591-f002:**
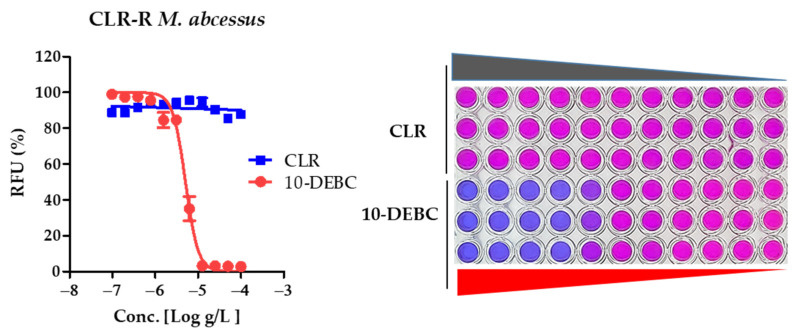
The activity of 10-DEBC against clarithromycin-resistant *M. abscessus* mutants. Clarithromycin-resistant *M. abscessus* 100 μg/mL to 97 μg/mL of CLR and 10-DEBC. DRC of *M. abscessus* CLR-R mutant (**Left panel**). Resazurin reports bacterial viability via a color change from blue to pink (**Right panel**). Upon oxidation by live *M. abscessus*, it turns pink, indicating growth of *M. abscessus*. Each experiment was performed in triplicate. This result was made from a representative experiment. The experiments were carried out with three biological replicates and expressed as the mean ± SEM for each concentration.

**Figure 3 ijms-23-00591-f003:**
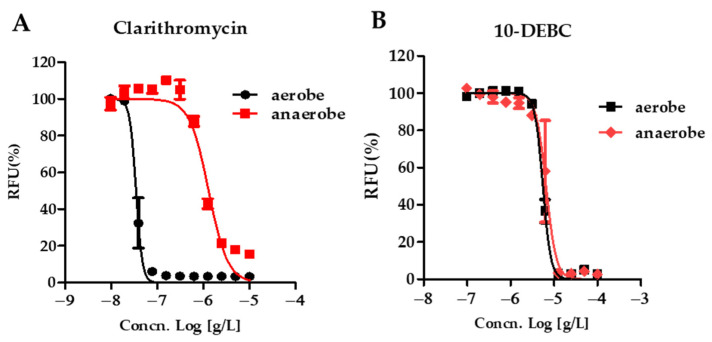
DRC of 10-DEBC. DRC of *M. abscessus* under aerobic, anaerobic using REMA. *M. abscessus* was undergone for seven days within an anaerobic generating container system (BD GasPak™ EZ). The activity of CLR (**A**) and 10-DEBC (**B**) against *M. abscessus* under aerobic and anaerobic conditions. The experiments were carried out with three biological replicates and expressed as the mean ± SEM for each concentration.

**Figure 4 ijms-23-00591-f004:**
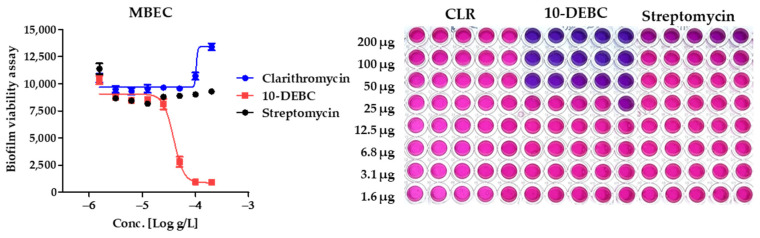
MBEC biofilm antimicrobial viability assay of 10-DEBC. The MBEC biofilm viability assay can confirm the effects of 10-DEBC-induced eradication on *M. abscessus* within biofilms by measuring the metabolic activities of live *M. abscessus* within biofilms using the resazurin. The experiments were carried out with three biological replicates and expressed as the mean ± SEM for each concentration.

**Figure 5 ijms-23-00591-f005:**
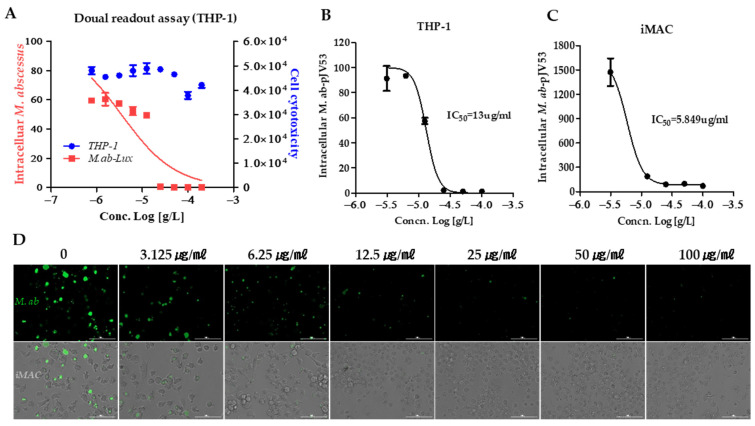
10-DEBC promotes the killing of intracellular *M. abscessus*. (**A**) Dual reading assay of 10-DEBC within macrophages. Infections in macrophages were carried out with multiplicities of infection (MOI) of 2 bacteria per cell for three hours and washed to remove extracellular mycobacteria. Luminescence detected from luciferase-expressing *M. abscessus* alone within THP-1 macrophages. The infected THP-1 cells stained with SYTO 60 were counted under fluorescence detection, and total cells were counted using Multi Reader. (**B**) THP-1 activated with PMA were infected at MOI of 2 with GFP-expressed *M. abscessus* for three hours, followed by treatment with 10-DEBC indicated concentrations in fresh medium. Ratio sum intensity values were used for all data reduction steps. (**C**) iMACs were infected at MOI of 2 with GFP-expressed *M. abscessus* for three hours, followed by treatment with 10-DEBC indicated concentrations in fresh medium. (**D**) Images of GFP-expressing *M. abscessus* infected iMACs on day three after treatment with indicated concentrations of 10-DEBC. Automated microscopy using a Lionheart^TM^ automated live-cell imager. The Gen5^TM^ 3.05 software object feature enables the identification of cells within the imaging field. Data were expressed as the mean ± SD of triplication for each concentration. (Scale bar = 100 μm.).

**Table 1 ijms-23-00591-t001:** The activity of 10-DEBC and antimicrobial agents against *M. abscessus* clinical isolates.

No.	Clinical Isolates	10-DEBC (IC_50_)	AMK (MIC)	CLR (MIC)	FOX (MIC)	Doxycycline (MIC)	Linezolid (MIC)	IPM (MIC)	Moxifloxacin (MIC)	Trimethoprim/ Sulfamethoxazole (MIC)
1	NCCP 13823(R)	1.86	16 (S)	1, >16 (IR)	32 (I)	>16 (R)	16 (I)	8 (I)	8 (R)	>8, 152 (R)
2	NCCP 13824(S)	1.74	16 (S)	1, >16 (IR)	64 (I)	>16 (R)	>32 (R)	8 (I)	>8 (R)	>8, 152 (R)
3	NCCP 13825(S)	1.78	16 (S)	8 (R)	128 (R)	>16 (R)	>32 (R)	32 (R)	>8 (R)	>8, 152 (R)
4	NCCP 13826(R)	1.77	16 (S)	2, >16 (R)	64 (I)	>16 (R)	32 (R)	16 (I)	>8 (R)	>8, 152 (R)
5	NCCP 13827(S)	2.44	8 (S)	0.25, 16 (IR)	64 (I)	>16 (R)	4 (S)	8 (I)	2 (I)	>8, 152 (R)
6	NCCP 13828(S)	2.63	8 (S)	1, >16 (IR)	32 (I)	>16 (R)	32 (R)	32 (R)	8 (R)	>8, 152 (R)
7	NCCP 13829(S)	1.69	16 (S)	1, >16 (IR)	32 (I)	>16 (R)	32 (R)	16 (I)	8 (R)	>8, 152 (R)
8	NCCP 13839(S)	1.86	8 (S)	2, >16 (R)	64 (I)	>16 (R)	32 (R)	16 (I)	>8 (R)	>8, 152 (R)
9	NCCP 15798(R)	2.24	8 (S)	0.25, 16 (IR)	64 (I)	>16 (R)	16 (I)	32 (R)	4 (R)	>8, 152 (R)

Abbreviation: susceptible (S), intermediate (I), inducible resistant (IR), resistant (R); antimicrobials and susceptibility breakpoints for non-tuberculous mycobacteria (NTM), as proposed by the Clinical & Laboratory Standards Institute (CLSI 2011) [22].

**Table 2 ijms-23-00591-t002:** Fraction unbound (*f_u_*) of CLR and 10-DEBC in caseum surrogate binding assay.

	*fu*	*fu* (%)	Undiluted *fu*
CLR	0.40 ± 0.01	39.92 ± 1.22	86.91 ± 0.58
10-DEBC	0.0060 ± 0.0005	0.60 ± 0.054	5.696 ± 0.501

All results are expressed as the mean ± standard deviation.

## Data Availability

Not applicable.

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
