# Peer review of "10-DEBC Hydrochloride as a Promising New Agent against Infection of Mycobacterium abscessus"

_ijms, 2022, doi:10.3390/ijms23020591_

Round 1

Reviewer 1 Report

Dear authors,

I found the topic of your research very important since there is a lack of effective drugs against Mycobacterium abscessus. I also appreciate the use of innovative assays like caseum surrogate binding assay. The research design is appropriate and well defined.

But there are some points that need to be addressed in my opinion.

Firstly, you must revise English language and be more careful in the writing of your object of research that is Mycobacterium abscessus, not abscesses or other spelling variants. I also found a phrase without the verb in the abstract (Efficacy of 10-DEBC without cytotoxicity in infected macrophages).

Secondly, you decided to perform MIC experiments, so I am puzzled, why do you decide to use IC50 as output of this kind of experiments? You should better use MIC90 or MIC99. This is the common output you obtain, and it is widely used in microbiology field.

Thirdly, your experiments are interesting, and you should find a way to better present them. In the Results section, it is not always very clear the aim of your experiments. For example, you tested the MIC against clarithromycin-resistant mutants, but you should write as first sentence that you decided to perform these experiments since the spread of clarithromycin resistance. Furthermore, you should also state clearer that these mutants are not clinical strains, maybe it would be better to have a separate paragraph.

After the careful revision of these points I raised, I think your manuscript will be improved since it is vital to present your data in the best possible way, this would also be beneficiary for the readers.

Author Response

I found the topic of your research very important since there is a lack of effective drugs against Mycobacterium abscessus. I also appreciate the use of innovative assays like caseum surrogate binding assay. The research design is appropriate and well defined.

But there are some points that need to be addressed in my opinion.

Firstly, you must revise English language and be more careful in the writing of your object of research that is Mycobacterium abscessus, not abscesses or other spelling variants. I also found a phrase without the verb in the abstract (Efficacy of 10-DEBC without cytotoxicity in infected macrophages).

>>>>>>> I have corrected all other spelling variants you pointed out.

Secondly, you decided to perform MIC experiments, so I am puzzled, why do you decide to use IC50 as output of this kind of experiments? You should better use MIC90 or MIC99. This is the common output you obtain, and it is widely used in microbiology field.

>>>>>>> I have added MIC90 values according to the reviewer’s comment in sections 2.1, 2.2, and 2.3.

Thirdly, your experiments are interesting, and you should find a way to better present them. In the Results section, it is not always very clear the aim of your experiments. For example, you tested the MIC against clarithromycin-resistant mutants, but you should write as first sentence that you decided to perform these experiments since the spread of clarithromycin resistance. Furthermore, you should also state clearer that these mutants are not clinical strains, maybe it would be better to have a separate paragraph.

>>>>> I have revised the sentence and explained clearly that these mutants are come from spontaneous experiments, not from clinical isolates, and separated the paragraph according to the reviewer's suggestion.

Reviewer 2 Report

10-DEBC hydrochloride as a Promising New Agent against Infection of Mycobacterium abscessus

Dear author and editor:

The resistance of NTM against antibiotic is a diffused problem with therapeutic difficulties. The article showed the antimycobacterial activity of 10-DEBC against Mycobacterium abscessus  and suggest it as a potential new drug candidate against Mycobacterium abscessus . the article could be published after a minor revision.

I have some comments on it:

  • Did you try to study the effect of 10-DEBC in combination with other antibiotics?
  • why you didn’t evaluate the efficacy of drug to reduce the intracellular CFU by counting the intracellular bacteria?
  • In the figure 5A and 5B it is better to write what Y axis refer to.
  • M. abscessus makes aggregate. for this reason, how did you count the single bacterium in figure 5?
  • What FM does mean in the text? Foaming macrophages ? please write it after the abbreviation?
  • The percent of 0.60% of fraction unbound. Does this percent significant?

Thank you very much, best regards

Author Response

The resistance of NTM against antibiotic is a diffused problem with therapeutic difficulties. The article showed the antimycobacterial activity of 10-DEBC against Mycobacterium abscessus and suggest it as a potential new drug candidate against Mycobacterium abscessus. the article could be published after a minor revision.

I have some comments on it:

Did you try to study the effect of 10-DEBC in combination with other antibiotics?

>>>>>>> Unfortunately, in this study, we did not try to combine 10-DEBC with other antibiotics. It’s a fascinating point. We will be sure to include it in future experiments.

why you didn’t evaluate the efficacy of drug to reduce the intracellular CFU by counting the intracellular bacteria?  M. abscessus makes aggregate. for this reason, how did you count the single bacterium in figure 5?

>>>>>>> We used the Bioluminescence (bacterial luciferase or bacterial green fluorescent), which is the production of light by luciferase-catalyzed reactions, is a versatile reporter technology with multiple applications both in vitro and in vivo. Bioluminescence has been applied to the study of infectious diseases, luminescent reporter strains have also been used for antibiotic testing in cells and permit the detection of microorganisms from within living animals, thus allowing the spatiotemporal study of infection in real-time in the same host. We considered the long duplication times of mycobacteria. Then we decided to utilize the advantages of bioluminescence as a reporter to assay anti-mycobacterial agents. The bioluminescence-based methods have proven more accurate than the traditional colony count methods. Another advantage is that it is possible to measure the amount of M. abscessus in consideration of the aggregation.

Although Bioluminescence is not possible to count M. abscessus one by one like CFU, it is the best way to measure the amount of M. abscessus in live macrophage cells in real-time.

In the figure 5A and 5B it is better to write what Y axis refer to.

>>>>>>> I insert the legend of the Y-axis in Figures 5A and 5B according to the reviewer’s suggestion.

What FM does mean in the text? Foaming macrophages? please write it after the abbreviation?

>>>>>>> I added the corresponding abbreviations in section 2.5.

The percent of 0.60% of fraction unbound. Does this percent significant?

>>>>>>> Yes this fu value (percentage) is pretty significant. The 10-DEBC’s average diluted fu value 0.06 % and undiluted fu value 5.696 are much higher than the bedaquiline, and clofazimine (fu ≦ 0.01 %).

Round 2

Reviewer 1 Report

The changed version of the paper followed the suggestions I made, thus it can be accepted in the present form.